# Synchronous deglacial thermocline and deep-water ventilation in the eastern equatorial Pacific

Natalie E. Umling[1] & Robert C. Thunell[1]

The deep ocean is most likely the primary source of the radiocarbon-depleted $CO_2$ released to the atmosphere during the last deglaciation. While there are well-documented millennial scale $\Delta^{14}C$ changes during the most recent deglaciation, most marine records lack the resolution needed to identify more rapid ventilation events. Furthermore, potential age model problems with marine $\Delta^{14}C$ records may obscure our understanding of the phase relationship between inter-ocean ventilation changes. Here we reconstruct changes in deep water and thermocline radiocarbon content over the last deglaciation in the eastern equatorial Pacific (EEP) using benthic and planktonic foraminiferal $^{14}C$. Our records demonstrate that ventilation of EEP thermocline and deep waters occurred synchronously during the last deglaciation. In addition, both gradual and rapid deglacial radiocarbon changes in these Pacific records are coeval with changes in the Atlantic records. This in-phase behaviour suggests that the Southern Ocean overturning was the dominant driver of changes in the Atlantic and Pacific ventilation during deglaciation.

[1] School of the Earth, Ocean and Environment, University of South Carolina, Columbia, South Carolina 29208, USA. Correspondence and requests for materials should be addressed to N.E.U. (email: numling@geol.sc.edu).

Atmospheric $CO_2$ plays an important role in setting Earth's thermostat and changes in atmospheric $CO_2$ provide the feedback mechanism needed to amplify the orbitally driven Pleistocene glacial–interglacial climate cycles[1]. Documenting the transfer of $CO_2$ between oceanic and atmospheric reservoirs is paramount to understanding the relationship between $CO_2$ and climate change. Radiocarbon studies are uniquely suited for identifying the pathway and rate of $CO_2$ exchange between the oceans and atmosphere by providing a tool for tracing carbon sources over the past 40,000 years. The last deglacial period (~20,000–10,000 years ago) is recent enough to be studied using radiocarbon and provides important insights into the nature of climate transitions and the role of tipping points in predicting abrupt shifts in the climate system[2].

Antarctic ice core records spanning the last deglaciation reveal a tight coupling between changes in atmospheric $CO_2$ and temperature[3]. In particular, these records contain two millennial scale episodes of increasing atmospheric $CO_2$ that are synchronous with decreases in atmospheric $\Delta^{14}C$ (refs 4,5). The first is a large 170‰ drop in $\Delta^{14}C$ and ~50 p.p.m.v. rise in $CO_2$ occurring during late Heinrich Stadial 1 (HS1; ~17.5–14.7 ka) and the second is a smaller $\Delta^{14}C$ drop of 70‰ and ~30 p.p.m.v. rise in $CO_2$ marking the end of the Younger Dryas[5,6] (YD; ~12.8–11.5 ka; Fig. 1). The magnitudes of these events suggest that the intermediate and/or deep ocean must be the primary source of this excess $CO_2$ (ref. 7), although identifying the specific source region has proven difficult. There is evidence to suggest that the large fluctuations in atmospheric $CO_2$ are driven by changes in Southern Ocean meridional overturning circulation[8].

Previous studies from both intermediate and deep water locations yield conflicting results regarding the magnitude and extent of the glacial carbon pool, as well as the timing of deglacial ventilation events[9–27], resulting in difficulty resolving both regional and inter-ocean ventilation signals. These inconsistencies may be due, at least in part, to the age models used for the different records. Traditionally, a constant reservoir age is used when calibrating foraminiferal radiocarbon dates[20,21,24]. However, variability in marine reservoir ages of at least several hundred years have been documented for the Pacific since the last glacial maximum from dating of contemporaneous marine and terrestrial samples[28], from correlation of marine tephra layers to dated terrestrial eruptions[22], and from paired U-Th and $^{14}C$

dating of corals[29]. In addition, reservoir age variability of >800 years has been suggested from tuning of plateaus in atmospheric $^{14}C$ (refs 30,31), from several tephra-dated records[19,27], and from correlation of climatic events to age control points in independently dated oxygen isotope records from the Hulu Cave speleothem or Greenland and Antarctic ice cores[13,17,25]. Without independent age control of radiocarbon-derived ventilation records it is difficult to quantify the actual magnitude of error introduced into initial foraminiferal radiocarbon content ($\Delta^{14}C_0$) calculations from age model uncertainty[32]. Furthermore, age model uncertainties make it difficult to determine whether ventilation events in different ocean basins occur synchronously or if there is an inter-ocean ventilation seesaw associated with the waxing of North Pacific Deep Water production when North Atlantic Deep Water production waned[24].

Here we reconstruct surface and deep water mass radiocarbon contents in the eastern equatorial Pacific (EEP) from sediment core TR163-23 (0°24′N, 92°09′W; 2,730 m water depth). This core site is located within the eastern equatorial cold tongue region off the Galapagos platform, along the return pathway of Pacific Deep Water (Fig. 2). The Galapagos region presently is influenced by the upwelling of nutrients and $CO_2$ from the Equatorial Undercurrent (EUC) into thermocline depth waters[33]. The southern hemisphere currently accounts for ~2/3 of the source waters feeding the EUC with Sub-Antarctic Mode Water (SAMW) formed along the northern edge of the Antarctic Circumpolar Current contributing much of the $\Delta^{14}C$ signature[33]. Consequently, TR163-23 is ideally positioned to record Southern Ocean-sourced ventilation of both thermocline and deep waters. We develop a well constrained age model for TR163-23 that does not require a static reservoir age and minimizes chronological uncertainty associated with the calculation of past radiocarbon content. Our records demonstrate that EEP thermocline and deep water ventilation occurred synchronously over the last 25,000 years. They also indicate that both Atlantic and Pacific ventilation were sensitive to changes in Southern Ocean deep-water formation consistent with a southern driver of deglacial climate change[34–36].

## Results

**Age model development.** To constrain age model uncertainty, three independent age models have been developed for TR163-23

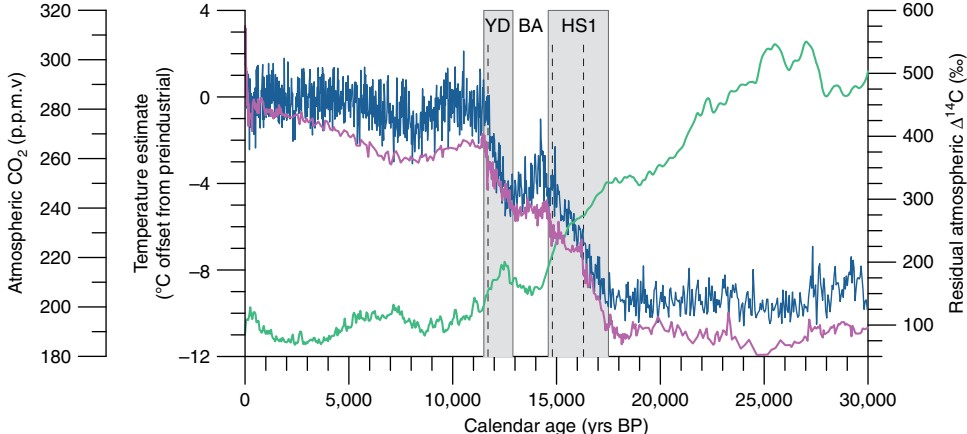

**Figure 1 | Atmospheric CO₂ and temperature changes across the last deglaciation.** Both atmospheric $CO_2$ from a composite record of Antarctic ice cores (blue)[65] and Antarctic temperature change recorded by the Epica Dome C ice core (pink)[3] show a two-step increase through the deglaciation. Increasing Atmospheric temperature and $CO_2$ during Heinrich Stadial 1 (HS1) and the Younger Dryas (YD) is separated by a period of little to no change during the BA. Atmospheric radiocarbon content ($\Delta^{14}C$ corrected for natural production; green)[6] decreases over the same period. Dashed lines indicate century-long periods of abrupt $CO_2$ rise identified by Marcott et al.[46]

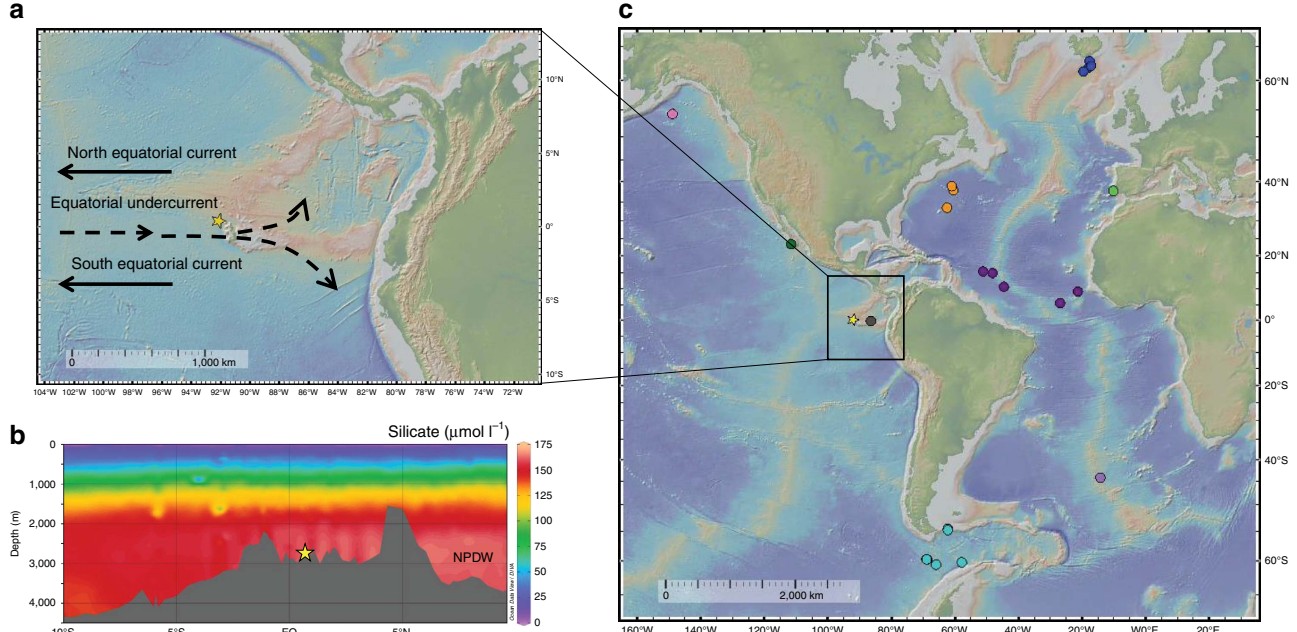

**Figure 2 | Core Locations and Study Area. (a)** Location of TR163–23 (0.41°N, 92.16°W; 2,730 m water depth) with the main bathymetric features and important currents of this region. **(b)** A cross-section of silicate concentrations along 88°W generated using the Ocean Data View software[66]; the location of TR163–23 is within the North Pacific Deep Water silicate maximum[58,67]. **(c)** The location of marine records displayed in Fig. 7 including TR163-23 (this study) shown with a star, ODP1240 (0.01°N, 86.46°W; 2,921 m)[18] in grey, MD02–2489 (54.39°N, 148.92°W; 3,640 m)[9] in pink, MV99-MC19/GC31/PC08 (23.47° N, 111.6°W; 705 m)[10,11] in dark green, Drake Passage corals[12] in light blue, MD07–3076 (44.07°S, 14.21°W; 3,770 m)[13] in light purple, a transect of S. Iceland cores (1,200–2,300 m)[14] in dark blue, northwest Atlantic corals[15] in orange, equatorial Atlantic corals[16] in dark purple, and eastern North Atlantic core MD99–2334k (37.8°N, 10.17°W; 3,146 m)[17] in light green.

(Fig. 3), along with an age model based on a constant reservoir correction ($\Delta R = 147 \pm 13$) of 31 *Neogloboquardina dutertrei* radiocarbon ages (Supplementary Tables 1 and 2). Two of the age models were derived from the tuning of a planktonic foraminiferal (*Globigerinoides ruber*) oxygen isotope record for TR163-23 (Supplementary Fig. 1) to the $\delta^{18}O$ records from the GRIP Greenland ice core (Fig. 3a)[37–39] and the Hulu Cave speleothems[40–42] (Fig. 3b). In addition, we used $^{14}C$ plateau tuning of the *N. dutertrei* radiocarbon dates following the procedure presented in Sarnthein *et al.*[30] (Fig. 3c) to develop a third independent age model. Reference record age estimates for the various tie points (Supplementary Table 3) were translated to TR163-23 using the elastic tie-point (ETP) method[43] and the final age models were developed using the Bayesian age-depth modelling program BACON[44] (Supplementary Tables 4 and 5). This method was chosen as it provides a computationally feasible way to quantify uncertainty associated with the transfer of age-depth estimates from a reference chronology. Tie-point constraint allows us to estimate reservoir age changes over the last glacial–interglacial transition and our sampling resolution of TR163-23 permits the observation of sub-millennial scale ventilation events. Above all, this technique of age model construction is not reliant upon the assumption of an unchanging reservoir age throughout deglaciation and provides more realistic estimates of age-model uncertainty. We did not have sufficient resolution during the Holocene to identify tie points and as a result this portion of the chronology was constructed using three calibrated ($\Delta R = 147 \pm 13$) *N. dutertrei* calendar ages. All three of the independently developed age models yield results that are indistinguishable from one another (Fig. 4). Conversely, the radiocarbon-based age model yields significantly different and generally older ages (Fig. 4). For the purpose of this study, we use the Hulu cave speleothem-tuned age model to interpret our results from the Galapagos. A more detailed description of the procedures used to establish each of the four age models is presented in the Methods.

**Thermocline and deep water ventilation.** The deep and thermocline carbon pools was assessed through calculation of the paleo-reservoir ages and the anomaly of the initial foraminiferal radiocarbon content ($\Delta^{14}C_0$) from the IntCal13 atmospheric record[45], or $\Delta^{14}C_{0\text{-atm}}$ as defined by Cook and Keigwin[32]. Contemporaneous atmosphere $^{14}C$ and $\Delta^{14}C$ values were averaged around associated Hulu-tuned age model $2\sigma$ error windows and the error was estimated from the standard deviation of each averaged value. Reservoir ages and $\Delta^{14}C_{0\text{-atm}}$ values were calculated from measured radiocarbon ages of mixed benthic foraminifera for deep waters and of *N. dutertrei* for thermocline depths (Fig. 5 and Supplementary Table 6). An initial increase in EEP $\Delta^{14}C_{0\text{-atm}}$ at $\sim 17.5$ ka coincides with the onset of rising atmospheric $CO_2$ at the beginning of HS1 (refs 3,46), with a second period of increasing $\Delta^{14}C_{0\text{-atm}}$ corresponding to the YD-Holocene transition (Fig. 6). These two episodes of gradually increasing $\Delta^{14}C_{0\text{-atm}}$ are separated by a period of generally well-ventilated thermocline and deep waters during the Bølling-Allerød (BA), a time when atmospheric $CO_2$ levels remained constant. For TR163-23, benthic and planktonic foraminiferal $^{14}C$ offsets from the atmosphere co-vary and as a result the Benthic-planktonic (B-P) offset changes very little over the past 22,000 years, remaining close to the modern difference of $\sim 1,250$ $^{14}C$ years (Fig. 5). Although the radiocarbon content in the EEP is primarily increasing throughout the deglaciation, there are three brief periods of little to no increase or even decline in

deep water $\Delta^{14}C_{0\text{-atm}}$ (15.6–15.1 ka; 14.6–13.3 ka; 13.1–12.4 ka). Each of these events is followed by a period of rapid $\Delta^{14}C_{0\text{-atm}}$ increase (15.1–14.6 ka; 13.3–13.1 ka; 12.4–10.8 ka). Our results clearly demonstrate that the $\Delta^{14}C_{0\text{-atm}}$ of EEP thermocline

and deep waters varied by at least 250‰ since the last glacial period (Fig. 6).

## Discussion

B-P $^{14}C$ offsets have often been used to estimate past ventilation changes[20,21,24]. However, concurrent increases in surface ocean reservoir ages and bottom water radiocarbon depletion, as seen in this study, may lead to the underestimation of ventilation changes calculated simply from B-P $^{14}C$ age differences[27]. While both TR163–23 and Panama Basin core ODP1240 (ref. 18) suggest similar B-P offsets during the deglacial, ODP1240 indicates an overall decrease of ∼870 years in the B-P offset from the glacial to the Holocene (Fig. 5). This dissimilarity in the two records could be due to differences in stratification through time between the two locations or it could be due to the fact that we have very little data for TR163–23 older than HS1 and thus are not capturing the true glacial B-P differences at our core location. Simultaneous deglacial benthic and planktonic reservoir age changes during deglaciation have also been observed in the southeast[22] and southwest Pacific[25,27,47]. This observation is consistent with modelling studies suggesting that surface reservoir ages are influenced not only by changes in lateral and vertical mixing, but also by the '$\rho CO_2$ effect' in which higher atmospheric $CO_2$ concentrations result in an increased oceanic equilibration rate of $\Delta^{14}C$ (ref. 48).

Our B-P and $\Delta^{14}C_{0\text{-atm}}$ results indicate that increases in both thermocline and deep-water radiocarbon content not only occurred synchronously over the last deglaciation but that the magnitude of change was similar at both depths (Fig. 6). Taken together, these records suggest that the initial oceanic 'flushing' of the respired carbon reservoir was likely completed by ∼14.5 ka and that any continued ventilation during the BA did not release a significant amount of carbon to the atmosphere. In addition, regrowth of the terrestrial biosphere during the BA may have contributed to the draw-down of $CO_2$ (refs 49,50), offsetting any oceanic $CO_2$ ventilation during this period. The $^{14}C$-depleted deep-waters at our site likely represent depletions within Pacific Deep Water as it flows southward at mid-water depths (2–3.5 km). Similar radiocarbon anomalies have been recorded at this depth range in high latitudes in the Gulf of Alaska[9], off New Zealand[26,27], in the northwest Atlantic[15] and in the south Atlantic[12,13]. Thus, this mid-depth carbon reservoir[23] appears to have been a pervasive feature of the glacial ocean. The ventilation pathway of this carbon to surface waters is less clear. The

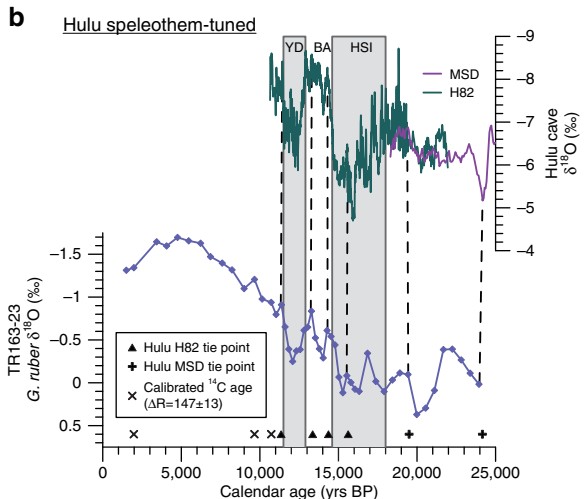

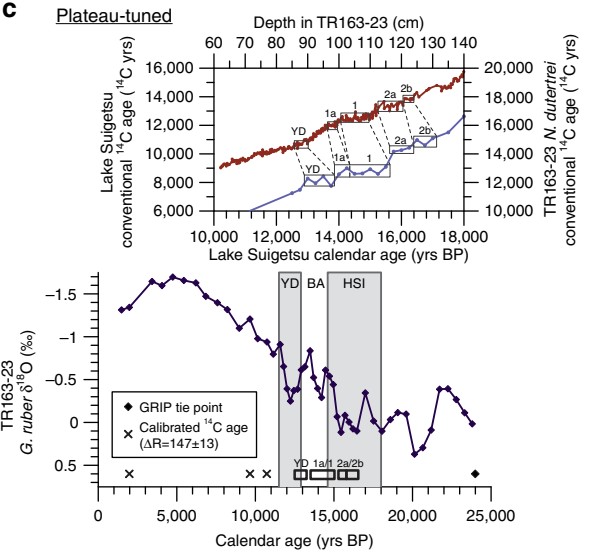

**Figure 3 | Construction of TR163-23 tuned chronologies.** (**a**) Tie point ages and age uncertainty for the TR163-23 G. ruber $\delta^{18}O$ record were constrained from the GRIP ice core $\delta^{18}O$ record[37–39] using ETP[43]. Calibrated radiocarbon ages used in the final age model are indicated with an X and ETP-derived tie-points are indicated with a diamond. (**b**) ETP constrained deglacial tie-points from the H82 speleothem and glacial tie-points from the MSD speleothem $\delta^{18}O$ records[40–42] are indicated by a triangle. As with the Greenland-tuned age model, calibrated radiocarbon ages are used to date the Holocene portion of the core in the final Bayesian-derived age model. Dashed lines connect TR163–23 $\delta^{18}O$ data points to corresponding Hulu speleothem and Greenland ice core reference record events used as tie points. (**c**) Five $^{14}C$-plateaus were identified in the TR163–23 N. dutertrei radiocarbon record (blue) that correspond to $^{14}C$-plateaus previously identified in the Lake Suigetsu atmospheric $^{14}C$ record (red)[63] by Sarnthein et al.[30]. A hiatus between the Younger Dryas plateau and plateau 1a was included in the final Bayesian-derived age model. In addition, a Greenland ice core tie point was used to constrain the LGM portion of the core. ETP, elastic tie-pointing; LGM, last glacial maximum.

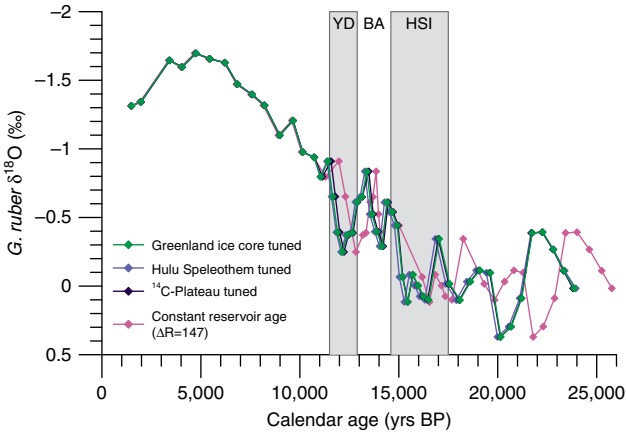

**Figure 4 | Comparison of age model techniques.** The three tuned age models yield results that are indistinguishable from one another within 1σ error windows. In contrast, the calibration of TR163–23 *N. dutertrei* [14]C-ages using a constant reservoir correction (ΔR = 147 ± 13) results in chronological offsets of up to ∼2,000 years from the tuned age models.

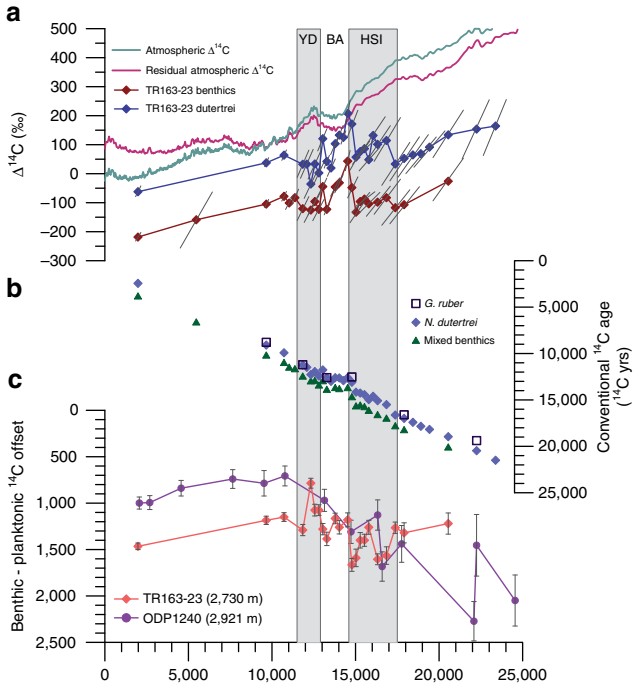

**Figure 5 | Radiocarbon records for TR163–23.** (**a**) TR163–23 benthic and planktonic $\Delta^{14}C_0$ and atmospheric records of $\Delta^{14}C$. The residual atmospheric $\Delta^{14}C$ has been corrected for modelled cosmogenic [14]C production[6]. Benthic and planktonic $\Delta^{14}C_0$ remains relatively constant through deglaciation except for a rapid increase at the end of HS1 and at the end of the BA. The major axis of 1σ error ellipses is plotted for benthic and planktonic $\Delta^{14}C_0$. (**b**) Conventional radiocarbon ages measured from 31 *N. dutertrei*, 3 *G. ruber*, and 25 mixed benthic samples. Deglacial *G. ruber* and *N. dutertrei* samples show negligible [14]C offsets. (**c**) Benthic-planktonic [14]C offsets in Galapagos core TR163–23 and Panama Basin core ODP 1240 (ref. 18) indicate similar offsets during the deglacial period. Although TR163–23 B-P offsets show little overall change from the LGM to the Holocene, ODP 1240 suggests a trend of decreasing B-P offsets during this period. One sigma error bars are plotted for B-P offsets. LGM, last glacial maximum.

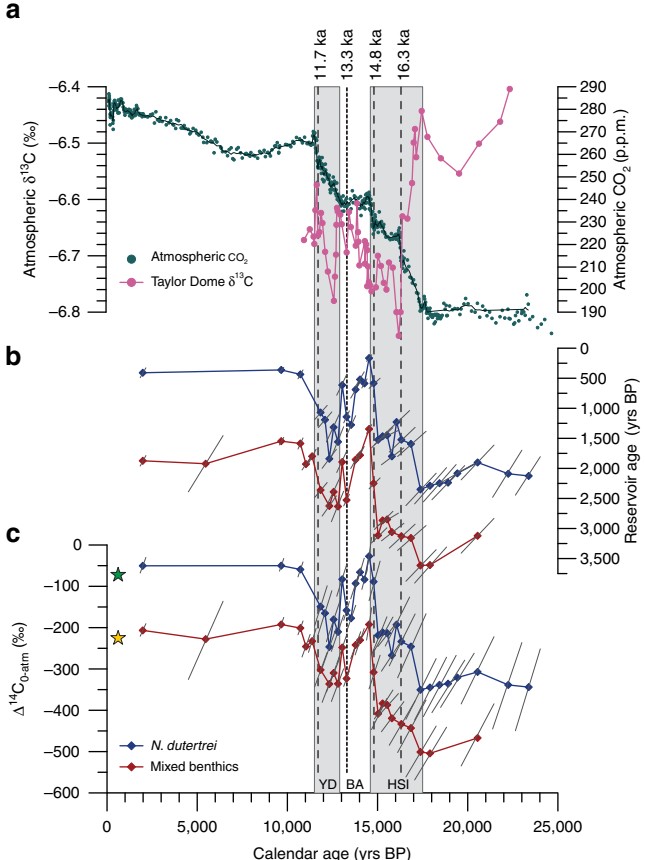

**Figure 6 | Foraminifera radiocarbon offsets from the atmosphere over the past 25,000 years.** (**a**) Composite atmospheric [CO$_2$] record from Antarctic ice cores[65] and $\Delta^{13}C$ from Taylor Glacier, Antarctica[49]. (**b**,**c**) Reconstructed radiocarbon offsets from the atmosphere displayed in calendar years and as $\Delta^{14}C_{O-atm}$ with the major axis of 1σ error ellipses displayed. Contemporary $\Delta^{14}C$ from GLODAP[58] is shown for thermocline depths (green star, −85‰) and for 2.7 km (yellow star, −220‰). Three centennial increases in the WDC atmospheric CO$_2$ record occur at 16.3, 14.8 and 11.7 ka (dashed line)[46]. There is an EEP equivalent at both thermocline and deep water depths for the 14.8 ka event, while a possible 16.3 ka event corresponds with only an increase in thermocline ventilation. Our records do not have sufficient resolution at 11.7 ka to capture ventilation changes associated with this century-long CO$_2$ increases. An additional ventilation event is recorded in TR163–23 in the late BA (dotted line). WDC, West Antarctic Ice Sheet Divide Ice Core.

simultaneous [14]C depletions in EEP deep and thermocline waters indicate that the EUC likely transported carbon to upwelling zones in the Eastern Pacific, much as it does at present. Indeed, boron isotope-derived pH estimates suggest that ventilation in the EEP was a significant source of CO$_2$ to the atmosphere during the deglaciation[51]. While SAMW is a major contributor to EUC waters, significantly aged SAMW has yet to be identified in the Southern Ocean after ∼18 ka (ref. 27). Although the north Pacific currently produces only ∼1/3 of EUC waters[52], it could have been a more important source of [14]C-depleted waters to the EUC during deglaciation.

As mentioned, the last deglaciation is marked by two millennial scale periods of increasing atmospheric CO$_2$ (17.5–14.5 ka; 13–11.5 ka), which are overprinted by several short-lived, rapid increases and plateaus[46] (Fig. 6). Generally,

**Table 1 | Method of age model constraint for cited radiocarbon records.**

| Archive | Region | Depth (m) | Age constraint | Reference |
|---|---|---|---|---|
| TR163–23 | Galapagos | 2,730 | Stratigraphic tie points (ice core $\delta^{18}O$ and $^{14}C$ plateaus) | This study |
| ODP1240 | Panama Basin | 2,921 | Stratigraphic tie points (correlation of Mg/Ca and $\delta^{18}O$ to Greenland ice core) | [18] |
| Coral transect | Equatorial Atlantic | 750–2,100 | Paired U/Th and $^{14}C$ ages | [16] |
| Coral transect | Drake Passage | 819–1,750 | Paired U/Th and $^{14}C$ ages | [12] |
| MD02–2489 | Gulf of Alaska | 3,640 | Stratigraphic tie points (ice core $\delta^{18}O$) | [9] |
| MD99–2334k | NE Atlantic | 3,146 | Stratigraphic tie points (correlation of Mg/Ca and $\delta^{18}O$ to Hulu cave speleothems) | [17] |
| MD07–3076 | Sub-Antarctic Atlantic | 3,770 | Stratigraphic tie points (Correlation of Mg/Ca to ice cores) | [13] |
| Iceland core compilation | N Atlantic | 1,237–2,303 | Correlation of ash layers to dated terrestrial eruptions | [14] |
| Coral compilation | NW Atlantic | 1,713–2,590 | Paired U/Th and $^{14}C$ ages | [15] |
| MV99-MC19/GC31/PC08 | E Pacific | 705 | Correlation of Diffuse Spectral Reflectance to GISP2 $\delta^{18}O$ | [10,11] |

Ventilation in TR163-23 was compared with records containing a chronology that allows for reservoir age variability.

each period of little to no increase or even decline in $\Delta^{14}C_{0\text{-atm}}$ appears to correspond with a pause in the build-up of atmospheric $CO_2$ (ref. 46) and may be indicative of the establishment of brief periods of stable climate conditions (Fig. 6). Alternating periods of oceanic ventilation and stratification during deglaciation appear to be tightly coupled to the increases and plateaus, respectively, in the $CO_2$ record, beginning with the initial rise in $CO_2$ and the onset of increasing ventilation in marine records at 17.5 ka (ref. 46). More importantly, this emphasizes the integral role of oceanic ventilation in modulating atmospheric $CO_2$.

The West Antarctic Ice Sheet Divide Ice Core atmospheric $CO_2$ record is marked by three centennial scale increases during the last deglaciation at ∼16.3, 14.8 and 11.7 ka (ref. 46 and Fig. 6). No marine equivalent of the 16.3 ka increase in atmospheric $CO_2$ has been identified in Southern Ocean $^{14}C$ records. However, high-latitude ventilation records from the North Pacific[9], Nordic Seas[53] and Iceland margin[14] record decreased deep water reservoir ages at ∼16 ka. Concurrent decreases in surface water reservoir ages have been recorded in the subtropical south Atlantic at 16.3 ka (ref. 31). While we see no evidence of North Pacific Deep Water ventilation at this time in our benthic $\Delta^{14}C_{0\text{-atm}}$ record, thermocline waters at our site display a small excursion towards younger radiocarbon values and less negative $\Delta^{14}C_{0\text{-atm}}$. This thermocline event is reduced in magnitude relative to that seen in North Pacific records and is consistent with a smaller proportion of northern relative to southern sourced waters incorporated into the EUC waters feeding our study site. Alternately, the 16.3 ka $CO_2$ increase may not be due to increased ocean ventilation but rather to the transfer of terrestrial carbon to the atmosphere[49,50]. In contrast, the 14.8 ka event at the HS1-Bølling transition is coincident with a ∼200‰ increase in $\Delta^{14}C_{0\text{-atm}}$ in both EEP thermocline and deep waters (Fig. 6). This is easily the largest magnitude ventilation change recorded in TR163–23 and it is likely responsible for release of a significant amount of the carbon stored in the mid-depth reservoir. After a period of generally decreasing $\Delta^{14}C_{0\text{-atm}}$ from ∼14.1 to 12.4 ka, TR163–23 records the resumption of increased EEP ventilation at ∼12.4 ka during the Younger Dryas. However, during this period our record has insufficient resolution to identify the presence of a distinct EEP ventilation event associated with the rapid 100–200 year long increase in $CO_2$ at ∼11.7 ka. Nevertheless, a significant rapid increase in $\Delta^{14}C_{0\text{-atm}}$ is recorded at this time in intermediate depth waters off Baja, CA, USA[10,11] supporting a link between ocean ventilation and the centennial scale $CO_2$ rise at ∼11.7 ka.

While the rapid increases in $CO_2$ at 14.8 and 11.7 ka coincide with changes in ocean ventilation, we also observe 75‰ and 100‰ $\Delta^{14}C_{0\text{-atm}}$ increases in EEP deep and thermocline depth waters, respectively, during the late-BA (∼13.0–13.3 ka) that are not associated with an increase in atmospheric $CO_2$ (Fig. 6). Although changes in thermocline depth or upwelling strength can result in $\Delta^{14}C$ variability, the magnitude of the late-BA increase in $\Delta^{14}C_{0\text{-atm}}$ far exceeds the range of thermocline driven seasonal and decadal variability observed in Galapagos coral radiocarbon over the past 400 years (ref. 54). This suggests that although upwelling may have increased at ∼13.3 ka, a change in endmember $\Delta^{14}C$ is required to produce an increase of the magnitude we observe at this time. Several explanations could account for the lack of an equivalent late-BA increase in atmospheric $CO_2$: (1) a more efficient biological pump reduced the 'leak' of $CO_2$ to the atmosphere; (2) the regrowth of terrestrial forests caused a draw-down of atmospheric $CO_2$; and/or (3) decreased air-sea exchange in the North Atlantic or Southern Ocean due to changes in wind stress and/or sea ice extent kept pace with ventilation, sequestering an amount of carbon comparable to that released in the EEP[46].

Using only marine $^{14}C$ records with independently constrained age models (Table 1) we find that EEP reservoir age changes are coherent with both shallow and deep Atlantic and Pacific reservoir ages, not only at the millennial scale but also with more rapid ventilation events (Fig. 7). Specifically, gradually increasing ventilation early in HS1 (∼17.5 ka) followed by an abrupt acceleration in the rate of ventilation coincident with the ∼14.8 ka rise in atmospheric $CO_2$ is recorded in the EEP (this study), the subtropical NE Pacific[10,11], the subpolar NE Pacific[9], the Equatorial Atlantic[16], the North Atlantic[14,15,17] and the subpolar South Atlantic[13]. Among these records, the EEP (this study), the subtropical NE Pacific[10,11], and the North Atlantic[14,17] also indicate a period of enhanced ventilation during the late Younger Dryas (Fig. 7). Using the Bayesian age-depth modelling program BACON[44,55], each of these records was analysed for the probability that the 14.8 ka ventilation event occurred at a given point in time from 13 to 18 ka. For each region, the probability of this event occurring at a given point in time was combined with the event probability for TR163–23 to determine the likelihood that EEP ventilation was occurring simultaneously with ventilation in the Atlantic and Southern oceans. Taking into account chronological uncertainty, rapid ventilation in the EEP during the HS1-BA transition was synchronous with the North Atlantic (Fig. 8) but did not show significant likelihood of synchronicity with the Southern Ocean.

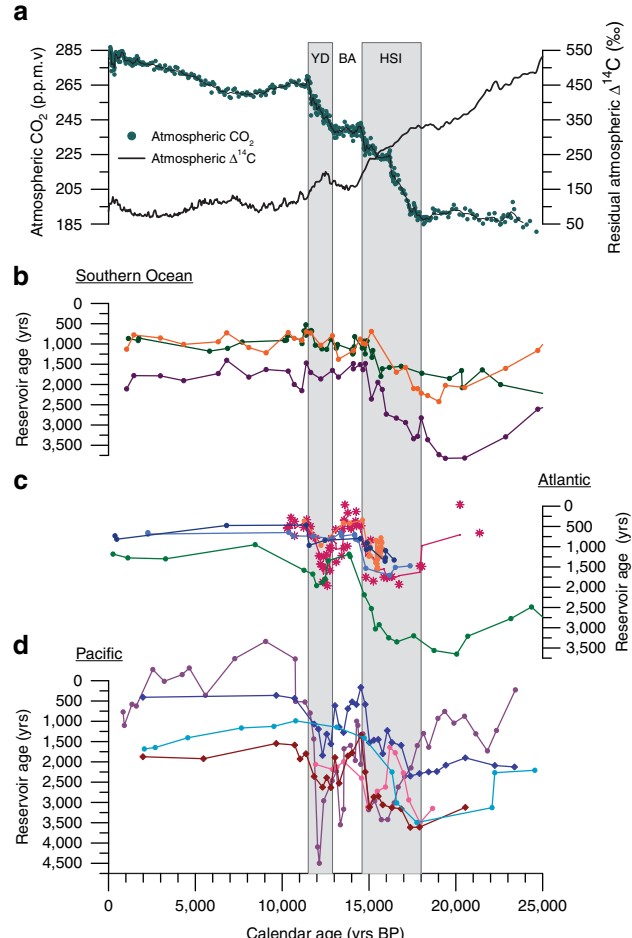

**Figure 7 | Inter-ocean comparison of shallow and deep water ventilation.**
(**a**) An Antarctic ice core compilation of Atmospheric $CO_2$ (ref. 65) indicates increasing $CO_2$ during the deglacial coeval with declining Atmospheric $\Delta^{14}C$ (corrected for natural production)[6,46]. (**b**) Southern Ocean records include surface (dark-orange) and deep (3,770 m; dark purple) water depths from core MD07–3076 (ref. 13) in the sub-Antarctic South Atlantic along with a Drake Passage coral compilation from Upper Circumpolar Deep Water[12] (dark green). (**c**) Atlantic records include deep-water core MD99–2334 k (ref. 17) (3,146 m; green) in the eastern Atlantic, a transect of cores off the southern coast of Iceland[14] (1,237–2,303 m; magenta), a coral compilation from the NW Atlantic[15] (819–1,750 m; orange), along with intermediate (1,296–1,492 m; navy-blue) and deep (1,827–2,100 m; light blue) water-depth equatorial Atlantic coral compilations[16]. (**d**) Pacific records include thermocline (dark blue) and deep (2,730 m; red) water depths from EEP core TR163–23 (this study), deep-water Panama Basin core ODP1240 (ref. 18) (2,921 m; cyan), deep-water Gulf of Alaska core MD02–2489 (ref. 9) (3,640 m; pink), and intermediate depth Eastern Pacific core MV99-MC19/GC31/PC08 (refs 10,11) (705 m; purple).

Rather, the Southern Ocean ventilation event associated with the 14.8 ka atmospheric $CO_2$ excursion[46] occurs slightly earlier than in the Eastern Pacific and the North Atlantic. Because of their low resolution, it is not possible to calculate the synchronicity of events from the equatorial Atlantic coral records.

The internal consistency of the Atlantic and Pacific ventilation records argues against the existence of inter-ocean differences in the timing of ventilation changes during the last deglaciation[24]. Furthermore, the early onset of ventilation in the Southern Ocean

due to overturning[36] suggests that this region was the dominant driver of deglacial ocean ventilation that then propagated northward into both the Pacific and Atlantic basins. The close correlation between Greenland ice core and Gulf of Alaska oxygen isotope records has been used as a basis for suggesting the close coupling and synchronization of poleward heat transport in the Pacific and Atlantic oceans during the last deglaciation[56] Similarly, the results of our study suggest that ventilation of the Atlantic and Pacific Oceans was also synchronous during the most recent deglaciation (Fig. 8).

## Methods

**Radiocarbon Analysis.** Sediment core TR163–23 (0°24′N, 92°09′W) was collected from 2,730 m water-depth off the western margin of the Galapagos Islands during R/V Trident cruise TR-163 in February 1975. Sedimentation rates in this core range from 8 to 12 cm kyr$^{-1}$ making it ideally suited for study of the last deglaciation. Samples of ~10 mg of the thermocline dwelling planktonic foraminifera *N. dutertrei* and 3–8 mg of mixed benthic species were measured for radiocarbon content at the University of California, Irvine Keck-Carbon Cycle AMS facility (Supplementary Table 1 and Supplementary Fig. 1). Mixed genus benthic foraminiferal samples were necessary to assure sufficient material for radiocarbon analysis. Deep infaunal species (*Globobulimina spp., Chilostomella oolina*) that may bias the results towards younger calendar ages were avoided[57]. The contemporary B-P offset was calculated using bomb corrected $\Delta^{14}C$ from the Global Ocean Data Analysis Project (GLODAP) bottle data[58], following De La Fuente *et al.*[18]. The modern $^{14}C$ ages were estimated from GLODAP $\Delta^{14}C$ using the equation from Stuiver and Polach[59] ($^{14}C$ age $= -8,033 \times \ln(1 + (D^{14}C/1,000))$ modified for $\Delta^{14}C$.

**Stratigraphic alignment.** In addition to the tuned age models, an age model was developed by applying a constant reservoir correction of 147 ± 13 years to 31 *N. dutertrei* radiocarbon dates using the Bayesian age-depth modelling program BACON[44]. Oxygen isotope analyses were carried out on the planktonic foraminifera *G. ruber* (Supplementary Fig. 1) using an Elementar Isoprime stable isotope ratio mass spectrometer. Distinctive oxygen isotope events were visually identified in TR163–23 and matched to Northern Hemisphere reference events in the GRIP Greenland ice core on the Greenland Ice Core Chronology 05 (GICC05)[37–39] for the Greenland-tuned chronology and to the H82 and MSD Hulu speleothems[40–42] for the Hulu-tuned age model. Tuning the EEP to Northern Hemisphere climatic events is supported mechanistically by an atmospheric bridge across the Isthmus of Panama[60]. During North Atlantic cold periods increased northeasterly winds drive down Panama drive decreased sea surface temperatures and shoaling of the thermocline resulting in an expansion of the equatorial cold tongue. The link between Northern Hemisphere climate change and the EEP has been recorded over the past 100 kyr by paleoclimate records of sea surface temperatures and sea surface salinity[61–63].

**Radiocarbon Plateau Tuning.** Visually identified TR163–23 radiocarbon plateaus were matched to Lake Suigetsu atmospheric $^{14}C$-plateaus[30,64] on the varve-derived Suigetsu age model consistent with published plateau-tuned age models[30,31]. This chronology is preferable to the modelled chronology because it is independent of assumed limits on changes in the dead carbon fraction of U/Th dated Hulu and Bahama speleothems[30]. Furthermore, the plateau-tuned age model developed using the varve chronology provides a closer match to the Hulu speleothem and Greenland ice core tuned age models for TR163–23. The indistinct base of plateau 1a/top of plateau 1 was not included in the final plateau-tuned age model. Plateau tuning suggest the presence of a short hiatus between the Younger Dryas plateau and plateau 1a. The timing and length of the hiatus is estimated using the Bayesian age modelling program BACON[44] with the prior distribution for the hiatus modelled by a gamma distribution with a mean chronological time gap of 1,000 years. Accumulation rates are assumed not to be auto-correlated between the depths just before and after the hiatus. However, inclusion of the hiatus has little effect on the final age model. The elastic tie-pointing method[43] was used to transfer tie-point ages from the reference record to TR163-23. Larger matching uncertainty was applied to the glacial tie-points in each of the tuned records to allow for a lower level of certainty associated with event contemporaneity during this period. Because we did not have sufficient resolution to identify tie-points during the Holocene this portion of the chronology was constructed using three calibrated ($\Delta R = 147 \pm 13$) *N. dutertrei* calendar ages for each of the tuned chronologies. In addition, radiocarbon plateaus could not be identified in TR163-23 during the glacial period requiring the use of one Greenland tie-point to constrain this period. Tie-point errors determined using the ETP method[43] and conventional radiocarbon ages were incorporated into the Bayesian age-depth modelling program BACON[44] to calibrate Holocene radiocarbon dates and to constrain the final age models and their uncertainty.

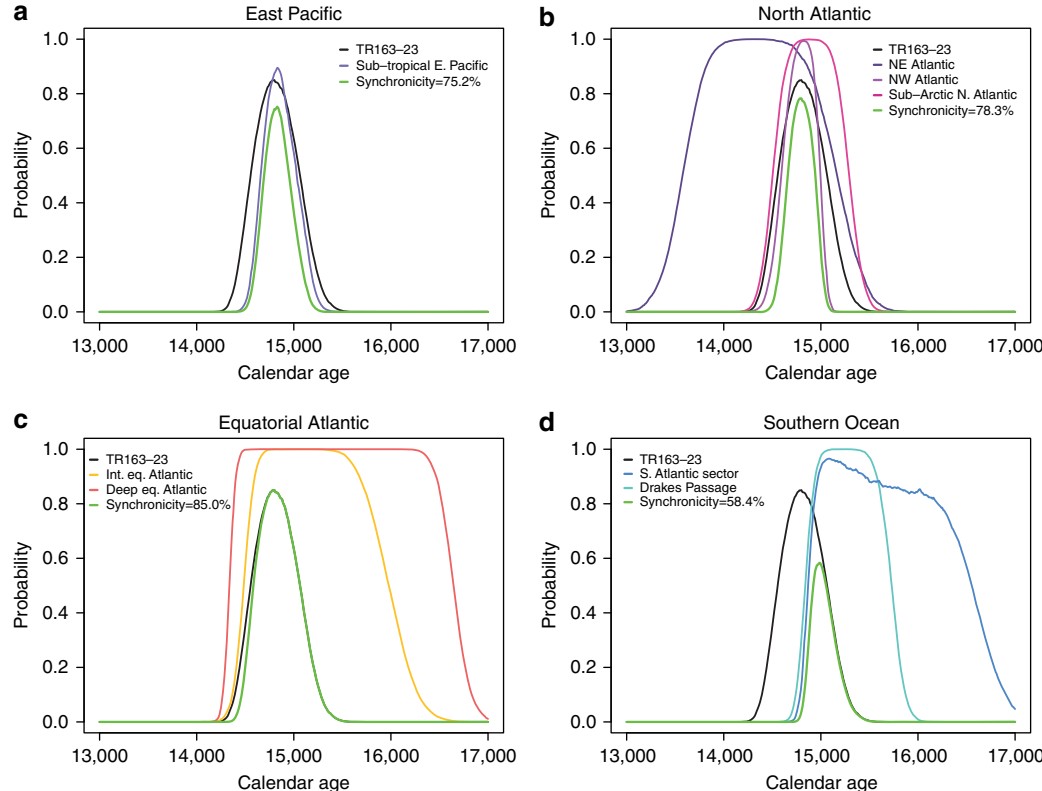

**Figure 8 | Inter-ocean synchronicity of the 14.8 ka ventilation event.** The Bayesian age modelling program BACON[44] was used to resample reported chronologies along a uniform time interval and to calculate the probability of the 14.8 ka ventilation event occurring at each study site from 13.0–17.0 ka using a 100-year window with a 10-year time step[55]. Incorporating age uncertainties, event probabilities were combined to test for synchronicity. The width of an individual probability curve is determined both by event length and age model uncertainty whereas the height is a function of age model certainty alone; for example, a narrow curve with peak probability close to 1.0 corresponds to a brief, well-defined event. Our EEP record is synchronous with the other (**a**) East Pacific (72.3%), and (**b**) North Atlantic (78.0%) records within 1σ (68.27%). (**c**) The 14.8 ka event in equatorial Atlantic records ends at a similar time as the EEP; however, this region has insufficient resolution at the onset of this event to determine whether the 14.8 ka event was indeed simultaneous with the EEP. (**d**) Southern Ocean records suggest an earlier ventilation event with less likelihood of synchronicity (48.7%) at 14.8 ka.

**Data Availability.** All data generated during this study are included in this published article and its Supplementary Information files.

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

## Acknowledgements

We thank E. Tappa for laboratory assistance and T. Heaton for proving the Elastic Tie-Pointing R code. We are grateful to J. Southon and the staff at the UC Irvine W.M. Keck Carbon Cycle AMC facility for providing AMS results. We also thank T. Guilderson for helpful comments that greatly improved the initial draft of the manuscript. We thank S. Carey from the University of Rhode Island Core Repository for providing the samples used in this study. Support for this curating facility is provided by the National Science foundation through Grant OCE-0956368. Partial support for this research was provided by a Geological Society of America Graduate Student Research Grant and a University of South Carolina SPARC Graduate Research Grant to N.E.U.

## Author contributions

R.C.T. initiated and guided the project. Data analysis and sample processing were carried out by N.E.U. Both N.E.U. and R.C.T. contributed to the data interpretation and preparation of the final manuscript.

## Additional information

**Competing financial interests**: The authors declare no competing financial interests.

**How to cite this article**: Umling, N. E. and Thunell R. C. Synchronous deglacial thermocline and deep-water ventilation in the eastern equatorial Pacific. *Nat. Commun.* **8,** 14203 doi: 10.1038/ncomms14203 (2017).

**Publisher's note**: 

