## [Peer Review File · Nature Communications]

Reviewers' comments:

Reviewer #1 (Remarks to the Author):

Umling and Thunell present new radiocarbon results reporting surface and deep ^{14}C age variability in the Eastern Equatorial Pacific (EEP) over the last glacial-interglacial transition. Through the obtention of a reasonably large set of radiocarbon dates (31 planktonic and 25 benthic samples) and the attempt to also establish past variations in the marine reservoir age, the authors conclude that the ventilation of deep waters and the thermocline occurred simultaneously and the magnitude of the change was similar at both depths. Although I think this manuscript represents a nice contribution to the increasing evidence of a glacial carbon pool, there are some issues that need to be better addressed or clarified.

The establishment of past reservoir age variability is crucial (although not necessarily straightforward) if we are to assess deglacial ocean ventilation. In this manuscript, the authors use two approaches: 1) correlation of ^{14}C 'plateaus' and 2) stratigraphic alignment to Greenland ice cores and Hulu Cave speleothems and conclude that both methods yield similar age models. This seems to be the case by looking at Figure 2, however, there is no information about how these age models were constructed, particularly the alignments with the $\delta^{18}\text{O}$ records from Hulu Cave and Greenland. How many tie-points were used? Where? Given that estimating reservoir ages is essential, the manuscript should include a more detailed information of the alignments and constraints used to establish the age models (supporting information and Fig.2). Also, regarding the ' ^{14}C plateau' approach, I wonder how robust the age model is if a more 'conservative' tuning is applied and some of the small plateaus are left out of the correlation.

Despite the assumptions made and the limitations of these 'tuning' approaches, the surface and deep reservoir ages depicted in Fig. 3 are in fairly reasonable agreement with previous studies from sites bathed in presumably the same water masses (e.g. Skinner et al, 2010). Yet, it is somehow surprising not to find a larger glacial ^{14}C depletion in deep waters compared to the thermocline (i.e. B-P differences are almost constant over the last 22 kyr, Fig. 3). Despite the EUC would have very likely brought 'very old' deep waters, upwelled near Antarctica, to the thermocline of the EEP during the deglaciation, a smaller ^{14}C drop at the thermocline still seems reasonable (despite the ' pCO_2 effect' on air-sea disequilibrium). Moreover, it is intriguing the different evolution of the B-P record when compared to the nearby core ODP1240 (de la Fuente et al., 2015). While the present study shows no ^{14}C offset between shallow and deep waters across the deglaciation, core 1240 shows a glacial/interglacial B-P change of approx. 900 years. Given the close proximity and similar water depths of the sites, I would expect similar shallow-deep ^{14}C gradients. Is there any explanation to account for these differences? It would be a good idea to include ODP1240 records in Figure 4 (surface and deep reservoir ages, which in general agree quite well) and further discuss these records in the manuscript. Maybe the $\delta^{13}\text{C}$ records of N. dutertrei could shed some light on this issue...

Section 'Atmospheric CO_2 -Ocean Ventilation Linkage' is a bit confusing and the brief periods/small changes discussed are sometimes difficult to identify. The age scale in Figure 3 does not help either. For instance, the four brief periods described in lines 167-169 are not obvious by looking at Figure 3 and I cannot see the correspondence with changes in atmospheric CO_2 . Also, in lines 192-194, 'After a period of generally increasing $\Delta^{14}\text{C}_{\text{O-atm}}$ from ~ 13.2 - 12.4 ka, TR163-23 records the resumption of increased EEP ventilation at ~ 12.4 ka during the Younger Dryas'. To me, from 13.5-12.5 ka there is a decrease, not an increase. I would recommend to focus on the three centennial changes depicted in the CO_2 ice core record. The resolution of the marine radiocarbon records does not allow to go much further.

Lines 239-244. 'The 13.6 ka period of reduced ventilation' is also confusing. It is hard to see from Figure 4 but I would say the period of reduced ventilation is centered at around 12.5 ka (\approx YD event). In any case, this is an interesting feature and clearly depicted in the EEP and the North

Atlantic (MD992334K, south of Iceland and corals, Fig. 4c), but not in the Southern Ocean (Fig. 4b), so it would not be 'geographically widespread'. In fact, this feature was previously described by Skinner et al, 2014 and justified as the result of a bipolar ventilation seesaw, accompanied by an increase in atmospheric CO₂ (as also seen in Figure 4).

Figures. I would recommend to add more tick marks and labels to the age scale axes. Also, legends with colors and name of the cores would be very useful in Figure 4, instead of or in addition to description in the figure caption.

Line 51, SOMOC (Southern Ocean meridional overturning circulation). This abbreviation is not necessary, it is not further used in the text.

Reviewer #2 (Remarks to the Author):

In this manuscript, the authors trace short-term changes in the deglacial transfer of CO₂ between oceanic and atmospheric reservoirs to reach a better understanding of the relationship between CO₂ and climate change. At a first glance one may assume that the manuscript largely follows similar studies recently published (e.g., Martinez-Botí et al. and de la Fuente et al., 2015). However, this study goes far beyond. Its main novelty lies in the generation of a highly precise age control that is crucial for any proper understanding of the actual phase relationships between changes in ocean ventilation and atmospheric CO₂ shifts.

To reach their goal the authors established detailed records of the radiocarbon (14C) reservoir ages of surface and deep-waters in the Eastern Equatorial Pacific, ages they found highly variable over the time span of the last deglaciation. Not only do reservoir ages reflect changes in ocean circulation, however, they are vital to any accurate age model construction for sediment core records that use radiocarbon tie points. The new record of surface water ages is based on a highly resolved planktonic 14C record evaluated by means of the 14C plateau tuning technique, where age-calibrated plateau boundaries provide a number of narrow spaced age control points. The records are derived from a sediment core with average sedimentation rates of 8-12 cm /kyr well suited to this study.

Specific comments:

1- The authors need to present a table showing the full set of planktonic and benthic radiocarbon ages, that includes the raw 14C ages vs core depth and (the finally derived) calendar age, the depth and age of planktonic plateau boundaries, the (accumulated) uncertainty ranges, resulting sedimentation rates, and all steps in the derivation of benthic reservoir ages, etc. etc. The present short note in Materials and Correspondence is unsatisfactory.

2- Fig. 2a shows that the present 14C sampling resolution is generally sufficient to distinguish most of 14C plateaus 1 and 2. However, the resolution may be easily improved in sediment sections near 80 and further up-core and near 128-140 cm depth. On this way some uncertainties of plateau tuning will be clearly reduced, most helpful for a general acceptance of this story (see below).

3- Plateau tuning: Assignment of the small plateau at 88-96 cm to the YD plateau appears controversial (Fig. 2a). The authors need to show their arguments why this plateau was not tuned to Plateau 1a. In case we follow the tuning proposed by the authors we miss an 800 yr long atmospheric 14C slope subsequent to ~12 cal. ka. Does this imply a small stratigraphic gap? Also,

a different assignment of the plateau now assigned to the YD will strongly reduce the high planktonic (and benthic) reservoir age of 1700 yr presently depicted in Fig. 3b.

Fig. 3 or 4a, and text: The authors should include a copy of the atmospheric $\delta^{13}\text{C}_{\text{CO}_2}$ record (Schmitt et al., 2012) to gain a better understanding of the mechanisms actually involved with the linkage of ocean ventilation and atmospheric CO_2 . For example (lines 199, 213, 222), I still see a problem to understand why and how the abrupt release of oceanic carbon actually has led to an increased contemporaneous ventilation by 'fresh' atmospheric ^{14}C in the surface ocean, that is, to strongly reduced reservoir ages of subsurface waters.

Minor concerns:

Fig. 5: Do the authors have any idea why the timing of the 14.8 ka event may occur that much later in the NE Atlantic than anywhere else? Is there a problem with the chronology of Skinner et al. (2014)?

Line 112: Insert more paragraphs for better legibility.

Line 148 - 149: Minor typos: of, depleted,

Line 221-222: typo? . . . increasing ventilation early in HS1 was followed by a rapid increase . . .

Further references:

Line 178: The authors regret to miss an equivalent of the 16.3 ka event in atmospheric CO_2 in the ^{14}C records of Atlantic surface waters. Incidentally I found a paper published last week in *Paleoceanography* that may help the authors to reproduce the 16.3 ka record of TR163-23 (Balmer et al.).

Line 263-264: Refer to Magana et al. (*Paleoceanogr.* 2010).

Resumé:

This well focused manuscript may become a milestone in the advance of climate research. It makes the noteworthy point that deglacial reservoir ages form a robust tracer of short-term changes in the carbon exchange between ocean and atmosphere. Understanding how surface reservoir age distributions change across the deglaciation is a topic of fundamental importance to the paleoclimatic community, thus requires urgent publication after minor revisions.

Reviewer #3 (Remarks to the Author):

Review of Synchronous Deglacial Shallow and Deep-Water Ventilation in the Eastern Equatorial Pacific for *Nature Communications*

The authors present a nice record of thermocline and deep (2730 m) radiocarbon from the last 25,000 years from the Eastern Equatorial Pacific from foraminifera, and use it to infer synchronous changes in ventilation in the Atlantic and Pacific. It is a very interesting record, and the paper is well written, however I am left with some questions and concerns, which leads me to suggest revisions before publication.

The first concern is the use of plateau tuning to construct an age model. This method has many hidden assumptions that are not well justified, and not clear to the average paleoceanographer. I suggest removing the plateau tuned age model, and recalculate the radiocarbon data using the

other age models. The two different age models from tuning with Greenland or Hulu, will provide the opportunity to test the sensitivity of the record to the assumed age model. I don't imagine it will make too much difference (given the similarities), but at least then the assumptions are obvious.

Plateau tuning makes the implicit assumption that a plateau in atmospheric radiocarbon (a decrease in $\Delta^{14}\text{C}$ of the atmosphere that happens faster than the decay rate) must occur at the same time as a rapid decrease in the surface ocean radiocarbon, and that it is not possible to generate a decrease in surface radiocarbon without changing the atmosphere similarly (or driving the change in the surface ocean by the atmosphere). While this assumption may be true at some times in the past, I don't think it has been shown that it has to be true at all times, and in all locations. For instance, as a thought experiment, consider a record of surface radiocarbon that experiences an increase in upwelling bringing radiocarbon depleted waters to the surface ocean. This would drive surface $\Delta^{14}\text{C}$ down, and generate a local radiocarbon plateau. However this might just be a local phenomenon that doesn't have any noticeable effect on atmospheric $\Delta^{14}\text{C}$, which would mean that there is a plateau in surface radiocarbon that is not at the same time as an atmospheric plateau, and thus aligning these records would introduce a bias in the age model. While not explicitly making an assumption of constant reservoir age, the plateau tuning method makes an implicit assumption that the surface reservoir age does not change dramatically during periods of rapid atmospheric $\Delta^{14}\text{C}$ fall. By aligning the plateaus in the surface and atmosphere records, you force the reservoir age over the plateau to be roughly constant over that interval, and you might be obscuring that by then calculating the reservoir age with a different atmospheric $\Delta^{14}\text{C}$ record (IntCal13) (see below). Thus the claim that the age model does not assume an unchanging reservoir age (line 93) is not quite accurate, since there is an implicit assumption of one at times.

Furthermore the use of the Suigetsu varve chronology as the atmospheric $\Delta^{14}\text{C}$ record for plateau tuning is not consistent with a number of things. First and foremost, it is not the preferred age model put forth by the authors of the Suigetsu radiocarbon paper. The varve chronology results in a lot of published marine radiocarbon data plotting above the atmosphere, which is not realistic, suggesting that that varve chronology is incorrect (as explained by the authors of that paper). Secondly, it is not consistent with the latest IntCal records, since IntCal13 takes into account the modeled age scale for the Suigetsu record. It is inconsistent to use the varve chronology for Suigetsu to come up with the timing of the plateaus and the age model of your core, and then to use IntCal13 to calculate radiocarbon offsets from the atmosphere, as you have essentially used two different atmospheric radiocarbon records to do your data processing, and they differ significantly.

My next major point revolves around Figure 5. This figure needs a lot more explanation, as it is not clear what went into these probability distribution functions, and how they were calculated. I'm particularly confused about what drives the width of the distribution. For instance for the Southern Ocean panel, the Drake Passage data has a window between about 14 ka to 16 ka, in contrast to the S Atlantic record which is much more narrow around 15. This is confusing because when you look at the radiocarbon data from figure 4 (Southern Ocean panel), the Drake Passage (green) shows an increase just before 15 ka, with it relatively flat before, whereas the South Atlantic (purple) shows a very gradual increase from about 20ka. I would think the window should be larger for the S Atlantic record compared to the Drake Passage, especially given the Drake Passage record has independent age control (uranium ages). Thus I don't really understand what is going into these figures and what is driving the shape of these histograms.

And finally a question – is there a difference in the conclusion of synchronicity between Atlantic and Pacific basins at different depths? In other words, the Atlantic-Pacific Seesaw (ref 27) was based on intermediate depth cores (1000 m) (albeit only two). Is your conclusion of synchronicity true for intermediate waters, or is it mainly a deep ocean conclusion? I can well imagine that the answers could be different, and that the upper ocean is responding differently than the deep ocean.

Minor points for revision:

The phrase floating reservoir (line 153, and 192) is not very clear, and should be avoided.

It would be good to plot the $\Delta^{14}\text{C}$ of the planktic and benthic radiocarbon, along with atmospheric $\Delta^{14}\text{C}$ so any changes in the inferred ventilation can be distinguished from changes driven by changing atmospheric $\Delta^{14}\text{C}$.

Figure 3 b would be clearer if the B-P record was plotted as a separate panel.

Reviewer #1

Comment #	Comments and response
1.1	“In this manuscript, the authors use two approaches: 1) correlation of ^{14}C 'plateaus' and 2) stratigraphic alignment to Greenland ice cores and Hulu Cave speleothems and conclude that both methods yield similar age models. This seems to be the case by looking at Figure 2, however, there is no information about how these age models were constructed, particularly the alignments with the $\delta^{18}\text{O}$ records from Hulu Cave and Greenland. How many tie-points were used? Where? Given that estimating reservoir ages is essential, the manuscript should include a more detailed information of the alignments and constraints used to established the age models (supporting information and Fig.2).” Response: We have added an additional figure (Figure 3) detailing the construction of each of the three tuned age models and elaborated on the techniques used (Methods). A table containing specific tie-points used for each age model is included in the supplementary materials (Supplementary Table 3).
1.2	“Also, regarding the '^{14}C plateau' approach, I wonder how robust the age model is if a more 'conservative' tuning is applied and some of the small plateaus are left out of the correlation.” Response: The Younger Dryas and 1a base/1 top plateau tie-points have been left out of the final plateau tuned age model. However, inclusion of these tie-points has little effect on the chronology. In addition, the age model produced from the plateau tuning technique is virtually identical to the other two tuned age models. Also, we now use the Greenland tuned age model as the primary age model for interpreting our results.
1.3	“Despite the assumptions made and the limitations of these 'tuning' approaches, the surface and deep reservoir ages depicted in Fig. 3 are in fairly reasonable agreement with previous studies from sites bathed in presumably the same water masses (e.g. Skinner et al, 2010). Yet, it is somehow surprising not to find a larger glacial ^{14}C depletion in deep waters compared to the thermocline (i.e. B-P differences are almost constant over the last 22 kyr, Fig. 3). Despite the EUC would have very likely brought 'very old' deep waters, upwelled near Antarctica, to the thermocline of the EEP during the deglaciation, a smaller ^{14}C drop at the thermocline still seems reasonable (despite the 'pCO_2 effect' on air-sea disequilibrium). Moreover, it is intriguing the different evolution of the B-P record when compared to the nearby core ODP1240 (de la Fuente et al., 2015). While the present study shows no ^{14}C offset between shallow and deep waters across the deglaciation, core 1240 shows a glacial/interglacial B-P change of approx. 900 years. Given the close proximity and similar water depths of the sites, I would expect similar shallow-deep ^{14}C gradients. Is there any explanation to account for these differences? “

	Response: We have added the ODP 1240 B-P record to Figure 5 and revised the text to include an explanation of the differences between ODP 1240 and TR163-23. In particular, we point out the B-P difference between the two core locations may be due to our lack of glacial samples older than HS1.
1.4	“It would be a good idea to include ODP1240 records in Figure 4 (surface and deep reservoir ages, which in general agree quite well) and further discuss these records in the manuscript. Maybe the d13C records of N. dutertrei could shed some light on this issue...” Response The ODP 1240 record of deep reservoir ages has been added to Figure 7 (old Figure 4).
1.5	“Section 'Atmospheric CO₂-Ocean Ventilation Linkage' is a bit confusing and the brief periods/small changes discussed are sometimes difficult to identify. The age scale in Figure 3 does not help either. For instance, the four brief periods described in lines 167-169 are not obvious by looking at Figure 3 and I cannot see the correspondence with changes in atmospheric CO₂. Also, in lines 192-194, 'After a period of generally increasing $\Delta^{14}C_{0-atm}$ from ~13.2-12.4 ka, TR163-23 records the resumption of increased EEP ventilation at ~12.4 ka during the Younger Dryas'. To me, from 13.5-12.5 ka there is a decrease, not an increase. I would recommend to focus on the three centennial changes depicted in the CO₂ ice core record. The resolution of the marine radiocarbon records does not allow to go much further.” Response: Line 192 ‘decreasing’ was a typographical error and was meant to read ‘increasing.’ we have corrected this error and for clarity we have added dashed lines to Figure 6 that indicate each of the centennial changes in the atmospheric CO₂ record along with a dotted line representing the additional ventilation event recorded in TR163-23 at 13.3 ka.
1.6	“Lines 239-244. 'The 13.6 ka period of reduced ventilation' is also confusing. It is hard to see from Figure 4 but I would say the period of reduced ventilation is centered at around 12.5 ka (\approx YD event). In any case, this is an interesting feature and clearly depicted in the EEP and the North Atlantic (MD992334K, south of Iceland and corals, Fig. 4c), but not in the Southern Ocean (Fig. 4b), so it would not be 'geographically widespread'. In fact, this feature was previously described by Skinner et al, 2014 and justified as the result of a bipolar ventilation seesaw, accompanied by an increase in atmospheric CO₂ (as also seen in Figure 4).” Response: TR163-23 records reduced ventilation at both 13.6 and 12.5 ka. We have revised the text to eliminate discussion of the 13. 6 ka period of reduced ventilation and instead refer to the ~13.3 ka ventilation event. In addition, the markers added to Figure 6 assist in eliminating the ambiguity when identifying referenced events.
1.7	“I would recommend to add more tick marks and labels to the age scale axes.” Response: The figures have been revised as suggested.

1.8	“Legends with colors and name of the cores would be very useful in Figure 4, instead of or in addition to description in the figure caption.” Response: Figure 7 (old Figure 4) has been revised as suggested.
1.9	“Line 51, SOMOC (Southern Ocean meridional overturning circulation). This abbreviation is not necessary, it is not further used in the text.” Response: The abbreviation has been removed.

Reviewer #2

Comment #	Comments and response
2.1	“The authors need to present a table showing the full set of planktonic and benthic radiocarbon ages, that includes the raw 14C ages vs core depth and (the finally derived) calendar age, the depth and age of planktonic plateau boundaries, the (accumulated) uncertainty ranges, resulting sedimentation rates, and all steps in the derivation of benthic reservoir ages, etc. etc. The present short note in Materials and Correspondence is unsatisfactory.” Response: We have included supplementary tables containing the radiocarbon data and age model information.
2.2	“Fig. 2a shows that the present 14C sampling resolution is generally sufficient to distinguish most of 14C plateaus 1 and 2. However, the resolution may be easily improved in sediment sections near 80 and further up-core and near 128-140 cm depth. On this way some uncertainties of plateau tuning will be clearly reduced, most helpful for a general acceptance of this story (see below).” Response: We have revised the manuscript to use the Greenland ice core tuned age model rather than the plateau tuned age model. However, the tuned age models are within error of each other.
2.3	“Assignment of the small plateau at 88-96 cm to the YD plateau appears controversial (Fig. 2a). The authors need to show their arguments why this plateau was not tuned to Plateau 1a. In case we follow the tuning proposed by the authors we miss an 800 yr long atmospheric 14C slope subsequent to ~12 cal. ka. Does this imply a small stratigraphic gap? Also, a different assignment of the plateau now assigned to the YD will strongly reduce the high planktonic (and benthic) reservoir age of 1700 yr presently depicted in Fig. 3b.” Response: Assigning the small plateau at 90-95 cm to plateau 1a requires a large change in sedimentation rate therefore we assign 90-95 cm to the Younger Dryas plateau and the top of plateau 1a to 100 cm. We do not include the small plateau at 90-95 cm in our final plateau tuned age model. Furthermore, we no longer use the plateau age model as the basis for interpreting our results.
2.4	“Fig. 3 or 4a, and text: The authors should include a copy of the atmospheric $\delta^{13}\text{C}\text{O}_2$ record (Schmitt et al., 2012) to gain a better understanding of the mechanisms actually involved with the linkage of ocean ventilation and atmospheric CO_2. For example (lines 199, 213, 222), I still see a problem to

	understand why and how the abrupt release of oceanic carbon actually has led to an increased contemporaneous ventilation by 'fresh' atmospheric ^{14}C in the surface ocean, that is, to strongly reduced reservoir ages of subsurface waters.” Response: We have included the atmospheric $\delta^{13}\text{CO}_2$ record in figure 6. We have also revised line 335 (old line 157) to clarify that increased ventilation in thermocline depth waters is responsible for the release of ^{14}C depleted CO_2 to the atmosphere.
2.5	“Fig. 5: Do the authors have any idea why the timing of the 14.8 ka event may occur that much later in the NE Atlantic than anywhere else? Is there a problem with the chronology of Skinner et al. (2014)?” Response: The NE Atlantic record from Skinner et al., (2014) does not have any samples from ~13.7-14.9 ka. Therefore, it is difficult to constrain the end of the ventilation event in this record resulting in higher event uncertainty. It is likely that the NE Atlantic event ends at a similar time as the other Atlantic records but the sampling resolution does not capture this.
2.6	“Line 112: Insert more paragraphs for better legibility.” Response: Text has been revised as suggested.
2.7	“Line 148 - 149: Minor typos: of, depleted,” Response: Typos have been revised.
2.8	“Line 221-222: typo? . . . increasing ventilation early in HS1 was followed by a rapid increase . . .” Response: Typos have been revised.
2.9	“Line 178: The authors regret to miss an equivalent of the 16.3 ka event in atmospheric CO_2 in the ^{14}C records of Atlantic surface waters. Incidentally I found a paper published last week in Paleooceanography that may help the authors to reproduce the 16.3 ka record of TR163-23 (Balmer et al.).” Response: We revised the text to include references to increased reservoir ages at ~16 ka in sub-polar North Atlantic deep waters (Thornalley et al., 2011; 2015) and sub-tropical Atlantic surface waters (Balmer et al., 2016) in addition to the North Pacific record previously referenced (Rae et al., 2014).
2.10	“Line 263-264: Refer to Magana et al. (Paleoceanogr. 2010).” Response: Reference has been added.

Reviewer #3

Comment #	Comments and response
3.1	“The first concern is the use of plateau tuning to construct an age model. This method has many hidden assumptions that are not well justified, and not clear

	to the average paleoceanographer. I suggest removing the plateau tuned age model, and recalculate the radiocarbon data using the other age models. The two different age models from tuning with Greenland or Hulu, will provide the opportunity to test the sensitivity of the record to the assumed age model. I don't imagine it will make too much difference (given the similarities), but at least then the assumptions are obvious.” Response: We have recalculated reservoir ages and $\Delta^{14}\text{C}_{0\text{-atm}}$ using the Greenland ice core tuned age model.
3.2	“Furthermore the use of the Suigetsu varve chronology as the atmospheric $\Delta^{14}\text{C}$ record for plateau tuning is not consistent with a number of things. First and foremost, it is not the preferred age model put forth by the authors of the Suigetsu radiocarbon paper. The varve chronology results in a lot of published marine radiocarbon data plotting above the atmosphere, which is not realistic, suggesting that that varve chronology is incorrect (as explained by the authors of that paper). Secondly, it is not consistent with the latest IntCal records, since IntCal13 takes into account the modeled age scale for the Suigetsu record. It is inconsistent to use the varve chronology for Suigetsu to come up with the timing of the plateaus and the age model of your core, and then to use IntCal13 to calculate radiocarbon offsets from the atmosphere, as you have essentially used two different atmospheric radiocarbon records to do your data processing, and they differ significantly.” Response: By recalculating reservoir ages and $\Delta^{14}\text{C}_{0\text{-atm}}$ using the Greenland ice core tuned age model rather than the Plateau tuned age model we avoid the inconsistencies associated with using both the Suigetsu and IntCal13 atmospheric radiocarbon records. However, when calculating the Plateau-tuned age model we chose to use the varve chronology consistent with published plateau-tuned age models (Sarthein et al., 2015; Balmer et al., 2016). This chronology is preferred to the modeled chronology because it is independent of assumed limits on changes in the dead carbon fraction of U/Th dated Hulu and Bahama speleothems (Sarthein et al., 2015). Furthermore, the plateau tuned age model developed using the varve chronology is consistent with both the Hulu speleothem and Greenland ice core tuned age models for TR163-23. To avoid the uncertainties associated with the Suigetsu chronologies we interpret our results using the Greenland ice core age model.
3.3	“My next major point revolves around Figure 5. This figure needs a lot more explanation, as it is not clear what went into these probability distribution functions, and how they were calculated. I'm particularly confused about what drives the width of the distribution. For instance for the Southern Ocean panel, the Drake Passage data has a window between about 14 ka to 16 ka, in contrast to the S Atlantic record which is much more narrow around 15. This is confusing because when you look at the radiocarbon data from figure 4 (Southern Ocean panel), the Drake Passage (green) shows an increase just before 15 ka, with it relatively flat before, whereas the South Atlantic (purple)

	shows a very gradual increase from about 20ka. I would think the window should be larger for the S Atlantic record compared to the Drake Passage, especially given the Drake Passage record has independent age control (uranium ages). Thus I don't really understand what is going into these figures and what is driving the shape of these histograms.” Response: In the figure 8 caption (old Figure 4) we have elaborated on the development of the probability curves and specified the factors driving curve width and height. Additionally, we have adjusted the definitions of the onset and end of each of the ventilation events in the S. Atlantic record and the two equatorial Atlantic records. These adjusted events allow for greater event uncertainty due to the lower sampling resolution of these three records.
3.4	“And finally a question – is there a difference in the conclusion of synchronicity between Atlantic and Pacific basins at different depths? In other words, the Atlantic-Pacific Seesaw (ref 27) was based on intermediate depth cores (1000 m) (albeit only two). Is your conclusion of synchronicity true for intermediate waters, or is it mainly a deep ocean conclusion?” Response: The ventilation event at ~14.8 ka is synchronous (88.7%) in the shallow N. Atlantic (Thornalley et al., 2011), intermediate E. Pacific (Marchitto et al., 2007; Lindsay et al., 2015), and a coral compilation from 1176-2441 m water depth (Robinson et al., 2005; Wilson et al., 2014). While more intermediate and shallow water-depth records are needed, it is likely that both intermediate and deep water-depth records are synchronous in the Atlantic and Pacific. However, deep water-depth records from the Southern Ocean hint towards an early onset of ventilation.
3.5	“The phrase floating reservoir (line 153, and 192) is not very clear, and should be avoided” Response: We have removed this phrase as suggested.
3.6	“It would be good to plot the $\Delta^{14}\text{C}$ of the planktic and benthic radiocarbon, along with atmospheric $\Delta^{14}\text{C}$ so any changes in the inferred ventilation can be distinguished from changes driven by changing atmospheric $\Delta^{14}\text{C}$.” Response: We have included the benthic and planktonic $\Delta^{14}\text{C}$ with the atmospheric $\Delta^{14}\text{C}$ in figure 5a.
3.7	“Figure 3 b would be clearer if the B-P record was plotted as a separate panel.” Response: We have placed the B-P record on a separate panel as suggested.

Reviewers' comments:

Reviewer #1 (Remarks to the Author):

I have read the new version of the manuscript, the comments of the other referees and the author's response to the reviews. One of the main concerns raised during the first round of reviews was related to the construction of age models (uncertainties of plateau tuning, selection criteria of tie-points,...). The authors now include an additional Figure and table with 14C data and age model information. I must say that I am impressed about the little difference, or virtually none, between the three age models (in fact, the original comparison (old Fig.2) wasn't as good as it is now). I have the impression that more 'realistic' age models will be obtained by using a more limited, and justified, number of tie-points. For instance, the tie-point around 20 kyr can be hardly correlated with a relevant event in Greenland and I am not very convinced either than the resolution of TR163-23 allows for the selection of two tie-points around 14 kyr (Greenland- and Hulu-tuned). To my view, a chronostratigraphic alignment with fewer constraints, despite it may decrease the precision of the age models, may be more credible and trustworthy.

Overall, I think the authors have done a good job trying to accommodate most of the comments of the referees. If the robustness of the stratigraphic alignment can be demonstrated by a better justified selection of tie-points, I would recommend this manuscript for publication in Nature Geoscience.

Other minor comments:

Line 21: I would avoid the term $\Delta^{14}\text{C}_0\text{-atm}$ in the abstract.

Line 50: The references go from n.11 to 20 and from n.22 to 30. What happened to ref. 21? Also, in the reference list, number 11 is missing.

Line 63: Define $\Delta^{14}\text{C}_0$. Now, the definition does not appear until line 118

Line 74 and 76. The reference should be 21 instead of 20.

Reviewer #2 (Remarks to the Author):

Review of a revised version of a manuscript of N.A. Umling and R. Thunnell entitled "Synchronous Deglacial and Deep-Water Ventilation in the Eastern Equatorial Atlantic" (Nature Comms. ms #: NCOMMS-16-14166-A)

In many respects the authors now met with care the reviewers' comments so that their basically highly valuable manuscript came closer to acceptance. The authors succeeded in improving the transparency of their data sets, results, and conclusions. In particular, the new Fig.3 and Tables 3 and 5 provide a fine comparison of two/three independent approaches of age control, thus have strengthened the plausibility of the results. Moreover, they clearly disprove the former concept of a long-term constant planktonic reservoir age (Table 2 and Fig. 4).

However, I still see a number of major and minor items, where the manuscript needs further clarifications and improvements and/or some basic reflections on the underlying reasoning of stratigraphic alignment to be finally accepted for publication.

(I) Figures 3A, B, and C still do not provide the necessary transparency about the original basis of the stratigraphic alignment of the planktonic d^{18}O record with the d^{18}O records of Greenland and Hulu Cave and the 14C plateau record. That is, it is necessary to present simple plots of the planktonic d^{18}O record vs. core depth instead of showing the tuned chronostratigraphic derivative of the record only. At present, a reader does not get a chance to, independently, assess and possibly reconcile the problems connected with the alignment of the d^{18}O record to the Younger Dryas and the problems of the 14C plateau near the base of the Younger Dryas (Comment #1.2).

(II) The tuning of the d18O record of *G. ruber* to GICC05 and/or Hulu is implicitly based on an important but hidden and fairly speculative assumption: The tuning implies that d18O changes of surface waters in the eastern equatorial Pacific are dominated by and synchronous with variations in northern hemisphere climate. However, the region of T163-23 (0°24'N) is marked by waters upwelled from the Equatorial Undercurrent, two thirds of which carry a Southern Hemisphere signal (see line 77-78). That means, the authors need to present the actual arguments why positive ('cold') excursions in d18O are strictly recording northern stadials such as the YD or, whether they rather reflect 'cold' events in the south such as the ACR, and/or possibly record a mixture of both northern and southern hemisphere signals.

To document these stratigraphic correlations, a figure that simply presents the planktic d18O record vs. core depth will be more objective, thus superior to the present plots of d18O vs. age. Also, the GICC05 tie points defined at more than 15 ka are difficult to verify in the d18O record of *G. ruber* in Fig. 3A, hence need to be marked more precisely and/or justified in the caption.

(III) Stimulated by comments of Reviewer #3 I have more closely inspected the details of plateau tuning, which lead me to recommend some minor and major additions/revisions:

1- The caption of the top panel of Fig. 3C needs to specify that the atmospheric 14C record of Suigetsu is plotted vs. calendar age and the planktonic 14C record of TR163-23 vs. core depth.

2- In Fig. 3C conjugate plateaus in the planktonic and atmospheric 14C records should be tied by thin guidelines. The lines will help to demonstrate that the YD plateau and Pl. #1 (possibly also including 1a) are separated in TR163-23 by a stratigraphic gap spanning 600, more likely 1000 yr, when the hiatus includes Pl. #1a.

3- **IMPORTANT:** The detection of a distinct stratigraphic gap (not esteemed by paleoceanographers) shows that plateau tuning may in part be superior to other age models independently applied in this manuscript, in contrast to a statement at the end of figure caption of Fig. 3 (line 489). That is, the effect of the plateau-based tie points is not 'negligible' and cannot be ignored by a Bayesian-derived final age model, since the outlined hiatus will unfortunately require minor-to-major changes in Figs. 4-7 and in the discussion of various events of deep-water ventilation during the upper B/A (see comment below on Fig. 6). To ignore distinct results of plateau tuning will necessarily lead to a biased age control, phase relationships, and conclusions not acceptable for publication.

4- The d18O record in the bottom panel of Fig. 3C should be plotted on a depth scale that enables an immediate comparison to the planktonic 14C record shown in the top panel right above. In this way we shall see that the hiatus between the base of the YD plateau and the top of Pl. #1 is matching a distinct vertical jump in d18O.

5- In contrast to the view of Reviewer #3 (Comment 3.2) the model-based ages for the atmospheric plateau boundaries of Suigetsu (Table 2, column 5) are likewise biased relative to varve-counted ages by up to 700 y near 17 ka due to restrictive model assumptions (e.g., a low variability of the Hulu Cave dead-carbon fraction; already listed by the authors). Accordingly, replacing them by the calendar ages directly deduced from varve counts, though also not perfect (Bronk Ramsey et al, 2015; Sarnthein et al., 2015), is a viable alternative, worth to be explored. In contrast to Reviewer's #3 plea for the model-based age scale the quality of science is not ruled by democratic votes but by the quality of scientific data and arguments, and the discussion is ongoing. By comparison to the model-based ages the ages based on varve counts lead to ages of the plateau boundaries that will come far closer to the age estimates derived from GICC05-tuning preferred by the authors (Table 2, column 2), in particular at the top and base of Plateaus 2a and b. Explanations in the table caption need to be completed accordingly.

6- I felt tempted to calculate the short-term changes in sedimentation rate between the various plateau boundaries presented in this manuscript and using the ages of Suigetsu varve counts, rates the changes of which provide indirect evidence of the quality of 14C plateau tuning. These rates generally turn out very constant, ranging from 15.6 cm/ky (YD Pl.), a 1000-y hiatus, up to 14.35 cm/ky for Pl.#1, 14.3 (base of Pl.#1 to top of Pl.#2), 9.6 for Pl. #2a, and 14.3 cm/ky for Pl.#2b. In case Pl.#1 is also including Pl.#1a, the joint interval will lead to rates of 9.8 cm/ky. In view of the high rates of most neighbor core sections this reduced value appears less likely, hence may support a long, that is a 1000 y hiatus from the top of Pl. #1 up to the base of the YD plateau. The outlined sedimentation rates should be displayed in a figure or table of the manuscript.

7- Also I calculated the reservoir age of *N. dutertrei*, averaged for each 14C plateau, and, where possible, for the surface dweller *G. ruber* (see attached figure). On average, these estimates come close to the reservoir ages listed in Table 5.

Table 2 caption does not specify the absolute value (547 y?) of the constant reservoir age employed.

(IV) (Fig. 6): I see a major threat to the meaning of the '13.3 ka event'. It may present a fake because of the 1000-yr (or 600-yr) long hiatus that probably spanned from 13050 (the base of YD 14C plateau) back to 14050 cal y (the top of 14C plateau #1) or at least back to 13640 cal y (the top of Pl. #1a). Further details on the results of plateau tuning were given above. - In general I see a problem in simply 'putting away' a major stratigraphic gap by means of the Bayesian-derived age model. Necessarily, the hiatus problem will also apply to the potential and possibly confusing (Comment 1.6; Lines 210-214) event of reduced ventilation now deduced for 13.6 ka, but actually located several hundred to a thousand years earlier.

(V) "Reservoir ages and D14C (0-atm) values were calculated from measured 14C ages of mixed benthic foraminifera for deep waters and of *N. dutertrei* for thermocline depths" (line 121-122). However, the authors still need to specify how they circumnavigate an old problem in their approach that suffers from two unknowns (in harmony with Adkins et al., 1997), (i) the actual provenance of glacial-to-deglacial deep waters in the Pacific and in particular, (ii) the surface water reservoir age at the site and time of deep-water formation that has not necessarily been the same as that measured for *N. dutertrei* at the site of foram deposition in the EEP.

In summary, I emphasize that the authors entered with their important observations a truly interesting field of science, which holds a key to our understanding of changes in the ocean, in a form finally suitable for publication in nature comms. However, I also admit, the authors entered a field full of 'booby traps' as seen by those, in part controversial, comments of the reviewers. For this reason, I suggest that the authors are urged for further revisions of their manuscript to reach an optimum transparency in the presentation of their data, their underlying hidden assumptions, and alternative approaches, and a most careful weighing of their conclusions.

Signed: Michael Sarnthein

Reviewer #1

Comment #	Comments and response
1.1	“I have the impression that more ‘realistic’ age models will be obtained by using a more limited, and justified, number of tie-points. For instance, the tie-point around 20 kyr can be hardly correlated with a relevant event in Greenland and I am not very convinced either than the resolution of TR163-23 allows for the selection of two tie-points around 14 kyr (Greenland- and Hulu-tuned). To my view, a chronostratigraphic alignment with fewer constraints, despite it may decrease the precision of the age models, may be more credible and trustworthy. Response: The tie points at ~20.0 ka, 14.1 ka, and 15.7 ka have been eliminated from the Greenland tuned age model and the tie points at ~14.1 ka and 17.0 have been eliminated from the Hulu tuned age model. Even with this more conservative tuning approach the age models remain within error. Because the precision of the Greenland age model did decrease we have decided to use the more precise but conservatively tuned Hulu age model to reduce the amount of error that is propagated into our $\Delta^{14}\text{C}_0$ calculations. The use of the Hulu tuned age model rather than the Greenland tuned age model has a negligible effect on our results. The tie points used for each of the age models are now more clearly identified in figure 3.
1.2	“Line 21: I would avoid the term $\Delta^{14}\text{C}_{0\text{-atm}}$ in the abstract. Line 63: Define $\Delta^{14}\text{C}_0$. Now, the definition does not appear until line 118.” Response: We have replaced the term $\Delta^{14}\text{C}_{0\text{-atm}}$ in the abstract and included a definition of $\Delta^{14}\text{C}_0$ earlier in the manuscript as suggested
1.3	“Line 50: The references go from n.11 to 20 and from n.22 to 30. What happened to ref. 21? Also, in the reference list, number 11 is missing. Line 74 and 76. The reference should be 21 instead of 20.” Response: The missing reference number 11 (Rae et al., 2014) has been added to the reference list and the numbering of references has been corrected.

Comment #	Comments and response
2.1	“Figures 3A, B, and C still do not provide the necessary transparency about the original basis of the stratigraphic alignment of the planktonic d18O record with the d18O records of Greenland and Hulu Cave and the 14C plateau record. That is, it is necessary to present simple plots of the planktic d18O record vs. core depth instead of showing the tuned chronostratigraphic derivate of the record only.” Response: We have included a plot of the planktic $\delta^{18}\text{O}$ vs. core depth in the supplementary information.
2.2	“The tuning of the d18O record of G. ruber to GICC05 and/or Hulu is implicitly based on an important but hidden and fairly speculative assumption: The tuning implies that d18O changes of surface waters in the eastern equatorial Pacific are dominated by and synchronous with variations in northern hemisphere climate. However, the region of T163-23 (0°24’N) is marked by waters upwelled from the Equatorial Undercurrent, two thirds of which carry a Southern Hemisphere signal (see line 77-78). That means, the authors need to present the actual arguments why positive (‘cold’) excursions in d18O are strictly recording northern stadials such as the YD or, whether they rather reflect ‘cold’ events in the south such as the ACR, and/or possibly record a mixture of both northern and southern hemisphere signals. To document these stratigraphic correlations, a figure that simply presents the planktic d18O record vs. core depth will be more objective, thus superior to the present plots of d18O vs. age.” Response: A Previous study by Xie et al (2008) has demonstrated a link between the Eastern Equatorial Pacific and northern hemisphere climate change driven by the atmospheric bridge across the Isthmus of Panama. Several studies have shown that this link drove SST variability not only during the deglaciation but also over the last 100,000 yrs (Kienast et al., 2006; Leduc et al., 2007; Dubois et al., 2014). The manuscript has been edited to reflect this link and a plot of the planktic $\delta^{18}\text{O}$ vs. core depth has been included in the supplementary information.
2.3	“Also, the GICC05 tie points defined at more than 15 ka are difficult to verify in the d18O record of G. ruber in Fig. 3A, hence need to be marked more precisely and/or justified in the caption.” Response: As mentioned above in response to comment 1.1, we have eliminated the Greenland tie points at 15.7 ka and 20.0 ka. In addition, dashed lines have been used to mark the tie point for the various age models with our $\delta^{18}\text{O}$ record.
2.4	“The caption of the top panel of Fig. 3C needs to specify that the atmospheric 14C record of Suigetsu is plotted vs. calendar age and the planktonic 14C record of TR163-23 vs. core depth.” Response: To avoid confusion figure axes in the top panel of figure 3c have

	been corrected to indicate that the age axis refers to the Suigetsu record and that the depth axis refers to the depth in TR163-23.
2.5	“In Fig. 3C conjugate plateaus in the planktonic and atmospheric 14C records should be tied by thin guidelines.” Response: Guidelines have been added to figure 3c, as well as figures 3a and 3b as requested.
2.6	“The detection of a distinct stratigraphic gap (not esteemed by paleoceanographers) shows that plateau tuning may in part be superior to other age models independently applied in this manuscript, in contrast to a statement at the end of figure caption of Fig. 3 (line 489). That is, the effect of the plateau-based tie points is not ‘negligible’ and cannot be ignored by a Bayesian-derived final age model, since the outlined hiatus will unfortunately require minor-to-major changes in Figs. 4-7 and in the discussion of various events of deep-water ventilation during the upper B/A (see comment below on Fig. 6). To ignore distinct results of plateau tuning will necessarily lead to a biased age control, phase relationships, and conclusions not acceptable for publication.” Response: The Bayesian analysis age modeling program BACON (Blauuw and Christen, 2010) allows for the inclusion of stratigraphic gaps. Using this method, the prior distribution for the hiatus is modelled by a gamma distribution with a mean chronological time gap of 1000 years. Accumulation rates are assumed not to be auto-correlated between the depths just before and after the hiatus. We ran several iterations of the plateau tuned age model to compare chronological differences related to inclusion of a hiatus and hiatus size.  1. The original plateau tuned age model as defined in the previous manuscript submission (no YD plateau). 2. An age model with the YD plateau (90-95 cm) included but without a hiatus. 3. A longer YD plateau as suggested in the figure supplied by reviewer 2 (90-97.5 cm). 4. Inclusion of a hiatus between the base of the YD plateau (97.5 cm) and the top of plateau 1a (100 cm). 5. Inclusion of a hiatus between the base of the YD plateau (97.5 cm) and the top of plateau 1 (100 cm). Plateau 1a is included in the hiatus. Despite inclusion of the hiatus, all iterations of the plateau tuned age model are within error of one another with the largest age model offsets occurring from ~12.5 ka to ~13.8 ka. In iteration 4 the hiatus is located from ~13.2-13.4 ka and in iteration 5 the hiatus is located from ~13.3-13.5 ka. Longer hiatuses result in a very large, rapid increase in sedimentation following the hiatus. We chose to use the plateau tuned age model containing a hiatus between the base of the YD plateau and the top of plateau 1a as this chronology resulted in the smoothest sedimentation rates and the best comparison to the $\delta^{18}\text{O}$ tuned age models.
2.7	“The d18O record in the bottom panel of Fig. 3C should be plotted on a depth

	scale that enables an immediate comparison to the planktonic ^{14}C record shown in the top panel right above. In this way we shall see that the hiatus between the base of the YD plateau and the top of Pl. #1 is matching a distinct vertical jump in $\delta^{18}\text{O}$.” Response: We have included a record of planktic $\delta^{18}\text{O}$ vs. core depth and ^{14}C vs. core depth in the supplementary information and chose to keep the x-axis in the bottom panel of figure 3c as age. This provides a direct comparison among the three age models presented in figure 3.
2.8	“By comparison to the model-based ages the ages based on varve counts lead to ages of the plateau boundaries that will come far closer to the age estimates derived from GICC05-tuning preferred by the authors (Table 2, column 2), in particular at the top and base of Plateaus 2a and b. Explanations in the table caption need to be completed accordingly.” Response: Use of the varve Suigetsu chronology for development of the plateau tuned age model along with its merits has been specified in the methods section.
2.9	“...the outlined sedimentation rates should be displayed in a figure or table of the manuscript.” Response: Sedimentation rates from each of the age models have been included in supplementary tables 2, 4, and 5.
2.10	“Table 2 caption does not specify the absolute value (547 y?) of the constant reservoir age employed.” Response: The reservoir correction applied to the constant reservoir age has been added to supplementary table 2.
2.11	“(Fig. 6): I see a major threat to the meaning of the ‘13.3 ka event’. It may present a fake because of the 1000-yr (or 600-yr) long hiatus that probably spanned from 13050 (the base of YD ^{14}C plateau) back to 14050 cal y (the top of ^{14}C plateau #1) or at least back to 13640 cal y (the top of Pl. #1a). Further details on the results of plateau tuning were given above. – In general I see a problem in simply ‘putting away’ a major stratigraphic gap by means of the Bayesian-derived age model. Necessarily, the hiatus problem will also apply to the potential and possibly confusing (Comment 1.6; Lines 210-214) event of reduced ventilation now deduced for 13.6 ka, but actually located several hundred to a thousand years earlier.” Response: As mentioned above in response to comment 2.6, the age model differences are within error of each other but the largest differences in age model occur from 12.5-13.8 ka. Also, as described in the response to comment 2.6, a small hiatus of ~200 years favors a smoother sedimentation rate than a larger hiatus of 1000 years. To reflect the uncertainty in hiatus timing and length, we eliminated the term ‘13.3 ka event’ in favor of a more conservative term, ‘late-BA event.’ Reference to the period of reduced ventilation at 13.6 ka

	was previously removed.
2.12	“... the authors still need to specify how they circumnavigate an old problem in their approach that suffers from two unknowns (in harmony with Adkins et al., 1997), (i) the actual provenance of glacial-to-deglacial deep waters in the Pacific and in particular, (ii) the surface water reservoir age at the site and time of deep-water formation that has not necessarily been the same as that measured for N. dutertrei at the site of foram deposition in the EEP.” Response: TR163-23 lies along the return pathway of Pacific Deep Water, predominantly fed by waters formed in the Southern Ocean. Tephra-tied and U/Th derived reservoir ages from depths >4 km in the Subpolar SW Pacific (Ronge et al., 2016) and S Atlantic (Burke et al., 2015) remain relatively constant through the last deglaciation. This suggests that changes in the surface reservoir age at sites of deep water formation in the Southern Ocean did not contribute significantly to the increase in EEP benthic reservoir ages if the Southern Ocean remained the dominant source of deep waters to TR163-23 throughout the deglaciation.

REVIEWERS' COMMENTS:

Reviewer #2 (Remarks to the Author):

In their point-by-point response letter and their manuscript the authors have satisfactorily addressed all points I had raised in my review, at least up to the cutting edge in our field of science, where all of us presently do not know any better answer /solution.

The manuscript is now appropriate for being accepted for publication in Nature Comm.